# Caveolin-Mediated Endocytosis: Bacterial Pathogen Exploitation and Host–Pathogen Interaction

**DOI:** 10.3390/cells14010002

**Published:** 2024-12-24

**Authors:** Dibyasri Barman, Rishi Drolia

**Affiliations:** Molecular and Cellular Microbiology Laboratory, Department of Biological Sciences, Old Dominion University, Norfolk, VA 23529, USA; dbarm001@odu.edu

**Keywords:** caveolae, endocytosis, host–pathogen interaction, caveolin-1, caveolin-2, caveolin-3, *Listeria monocytogenes*, microbial invasion, immune evasion

## Abstract

Within mammalian cells, diverse endocytic mechanisms, including phagocytosis, pinocytosis, and receptor-mediated endocytosis, serve as gateways exploited by many bacterial pathogens and toxins. Among these, caveolae-mediated endocytosis is characterized by lipid-rich caveolae and dimeric caveolin proteins. Caveolae are specialized microdomains on cell surfaces that impact cell signaling. Caveolin proteins facilitate the creation of caveolae and have three members in vertebrates: caveolin-1, caveolin-2, and caveolin-3. Many bacterial pathogens hijack caveolin machinery to invade host cells. For example, the Gram-positive facultative model intracellular bacterial pathogen *Listeria monocytogenes* exploits caveolin-mediated endocytosis for efficient cellular entry, translocation across the intestinal barrier, and cell–cell spread. Caveolin facilitates the internalization of group A streptococci by promoting the formation of invaginations in the plasma membrane and avoiding fusion with lysosomes, thereby aiding intracellular survival. Caveolin plays a crucial role in internalizing and modulation of host immune responses by Gram-negative bacterial pathogens, such as *Escherichia coli* K1, *Klebsiella pneumoniae*, *Pseudomonas aeruginosa*, and *Salmonella enterica* serovar Typhimurium. Here, we summarize how bacterial pathogens manipulate the host’s caveolin system to facilitate bacterial entry and movement within and between host cells, to support intracellular survival, to evade immune responses, and to trigger inflammation. This knowledge enhances the intervention of new therapeutic targets against caveolin in microbial invasion and immune evasion processes.

## 1. Introduction

The plasma membrane is not just a boundary for the cell; it is an active, multi-functional system involved in many cellular processes. In various types of cells, the plasma membrane is covered with small pits, 50–80 nm in size, known as caveolae, which form specialized regions within the membrane and were initially identified by Palade and Yamada in the 1950s [1]. These plasma membrane domains are principally formed by caveolins, which are essential for the biogenesis of caveolae [2,3,4]. The first caveolin protein was discovered in 1989 by Glenney and colleagues [5,6,7]. Caveolins are oligomeric, cholesterol-binding integral plasma membrane proteins, ranging from 17 to 24 kDa, and are crucial in the invagination process of caveolae from the plasma membrane. To form caveolae, a core set of structural components is required, including caveolin-1 (Cav-1), caveolin-2 (Cav-2), caveolin-3 (Cav-3), cavin-1, and PACSIN/Syndapin proteins [8]. Following the discovery of Cav-2 and Cav-3, the original caveolin was renamed Cav-1 [9,10]. Cav-2 co-localizes with Cav-1 and is expressed in a variety of tissues, with the highest levels found in endothelial cells, adipocytes, fibroblasts, and smooth muscle cells, whereas Cav-3 expression is restricted to skeletal and cardiac muscle [7]. Recent cryo-electron microscopy studies suggest that the human Cav-1 complex is composed of 11 protomers organized into a tightly packed disc with a flat membrane-embedded surface [11].

The stability of caveolae also depends significantly on Eps15 homology domain (EHD) proteins, which associate with the neck region of caveolae and are essential for both caveolae formation and trafficking [8,12,13,14]. Cav-1 induces membrane curvature, an essential feature for creating caveolae [8], while cavins, which are peripheral membrane proteins with lipid-binding capabilities, form oligomeric complexes to stabilize caveolae structure [8,11,15]. Cavin1, for instance, binds both phosphatidylinositol 4,5-bisphosphate (PtdIns(4,5)P2) and phosphatidylserine (PtdSer), further supporting the stability of Cav-1-rich areas [16,17]. Knockdown of PACSIN/Syndapin proteins revealed that the absence of these proteins led to the loss of caveolae morphology, suggesting their crucial role in membrane deformation required for caveola generation [8]. Beyond caveolae formation, PACSINs also contribute to caveolae function by recruiting dynamin-II to caveolae and interacting with EHD proteins [12].

Caveolins are thought to adopt a hairpin-like shape within the membrane, with their C- and N-termini facing the cytoplasm [8]. They are synthesized as integral membrane proteins in the endoplasmic reticulum (ER) via a signal recognition particle (SRP)-dependent pathway. Caveolins are transported through the Golgi apparatus to the plasma membrane [18]. They combine with cavin-1 to form caveolae [19,20,21]. At the plasma membrane, caveolae represent a unique type of lipid raft with caveolin proteins, diverse lipid species, an invaginated structure, and enriched lipid leaflets. Cholesterol is essential for assembling these caveolae, along with high concentrations of negatively charged phospholipids, such as PtdSer and PtdIns(4,5)P2, which contribute to a distinct lipid environment within the caveolae [16].

Endocytosis is a cellular uptake process by which the cell regulates the translocation of molecules from outside. Based on the mechanism of endocytosis, it is subdivided into various categories. At the whole cell level, it is sub-divided into phagocytosis and pinocytosis. Phagocytosis is the engulfment and elimination of large particles by phagocytic cells such as macrophages and monocytes. On the other hand, pinocytosis is the uptake of small particles by non-phagocytic cells. Based on the dependency of the tetrameric GTPase dynamin (specifically dynamin-II), pinocytosis is again subdivided into two groups. The pinocytosis requiring dynamin-II is subdivided into clathrin-mediated endocytosis and caveolin-dependent endocytosis. Whereas dynamin-independent pathways are sub-divided into lipid-raft-dependent endocytosis, flotillin-dependent endocytosis, and macropinocytosis [22].

Caveolin-mediated endocytosis begins with the internalization of cargo through caveolae, which are specialized membrane invaginations rich in cholesterol and sphingolipids. Dynamin-2, a crucial GTPase, is essential for the fission of these caveolae from the plasma membrane. Following internalization, caveolin-positive vesicles may interact with early endosomes, where initial sorting occurs. Depending on the specific cargo and cellular needs, the vesicles can diverge to various intracellular compartments, including the Golgi apparatus or signaling endosomes. Unlike clathrin-mediated endocytosis, caveolin-mediated pathways often bypass lysosomes, avoiding degradation and instead supporting roles in signaling, lipid regulation, and transcytosis [8,23].

Caveolae is crucial in endocytosis, cholesterol and lipid metabolism, mechanosensation, and cellular signaling [7]. Alongside these major functions, several bacterial pathogens hijack caveolin- and caveolae-mediated pathways to bypass traditional host defenses, allowing them to establish infections and evade the immune system [24,25,26]. Over the past 70 years since its discovery, caveolin-mediated endocytosis and caveolin proteins have emerged as a central theme for their role in bacterial pathogenesis. They are now increasingly recognized as an essential factor in how bacteria invade, survive, and manipulate host cells [25,27,28,29]. In this review, we comprehensively summarize past and recent discoveries on how various bacterial pathogens exploit the host’s caveolin machinery for bacterial internalization, translocation into and across host cells, intracellular survival, immune evasion, and inflammation.

## 2. Gram-Positive Pathogens That Hijack the Caveolin-Mediated Endocytosis for Cellular Entry, Survival, and Immune Evasion

### 2.1. Listeria monocytogenes

*L. monocytogenes* is a model Gram-positive facultative intracellular bacterial foodborne pathogen that causes listeriosis. Among the foodborne pathogens, it ranks third in the deaths caused yearly due to a high hospitalization (94%) and a high mortality rate (20–30%) [30]. The bacterium exploits its ability to cross critical host barriers, such as the intestinal, blood–brain, and placental barriers, which is crucial in its infection strategy [31,32]. Invasion and translocation across the gut epithelial cells are dependent on two essential proteins: Internalin A (InlA) and the *Listeria* adhesion protein (LAP) [33,34,35]. While the entry of *L. monocytogenes* via Peyer’s patches is mediated by the surface protein InlB [36]. After entry, the bacterium becomes enclosed in a phagosome, which is degraded by listeriolysin O (LLO), a protein encoded by the *hly* gene, allowing it to access the cytosol [37]. Once in the cytosol, *L. monocytogenes* employs the ActA protein to recruit the Arp2/3 complex, driving actin polymerization to enable its spread from one cell to another [38]. The formation of protrusions by *L. monocytogenes* also requires the secreted bacterial protein InlC, which interacts with the SH3 domain at the carboxyl terminus of the human scaffolding protein Tuba [39,40].

To cross the gut intestinal epithelial cells and traverse the intestinal barrier, *L. monocytogenes* utilizes the coordinated action of LAP-dependent paracellular translocation and InlA-mediated transcytosis [35]. The bacterial surface protein LAP binds to its host cell surface receptor heat shock protein 60 (Hsp60), leading to the internalization of tight junction proteins, claudin-1, occludin, and the adherens junction protein E-cadherin via Cav-1 and MLCK-mediated endocytosis [33,35,41,42] (Figure 1). Recent studies by our group have shown that pharmacological inhibition of caveolin in cell lines (Caco-2) and genetic knockout of Cav-1 in mice blocks LAP-induced increases in intestinal permeability, junctional endocytosis, and *L. monocytogenes* translocation [43]. Once internalized, these junctional proteins are processed in early and recycling endosomes, resulting in the opening of cell junctions that permit *L. monocytogenes* to navigate through the intercellular spaces [43]. Following this initial junction opening by LAP, InlA subsequently binds directly to its receptor E-cadherin, facilitating InlA-mediated transcytosis across the gut epithelial barrier [43] (Figure 1).

While pioneering studies by Veiga et al. highlighted the essential role of clathrin in the initial actin-based invasion of *L. monocytogenes* into host cells, thereby establishing “bacterial-induced invasion” as the function of this nanoscale endocytic pathway, the role of caveolae in *L. monocytogenes* uptake could not be ruled out [44]. In cultured non-phagocytic cells, *L. monocytogenes* utilizes its surface proteins, InlA and InlB, to interact with human E-cadherin (hEcad) and c-Met, respectively, leading to cytoskeletal rearrangement via a zipper mechanism. Depleting cholesterol with methyl-β-cyclodextrin (MβCD) reduced the internalization of *L. monocytogenes* via both internalins, indicating the role of lipid rafts in this entry mechanism [45]. Various lipid raft markers, including glycosylphosphatidylinositol-linked proteins and the ganglioside GM1, were observed at the entry sites, supporting the involvement of lipid rafts in internalization. While the interaction between InlA and E-cadherin is cholesterol-dependent, cholesterol depletion does not affect the InlB-c-Met interaction or its downstream phosphoinositide-3 kinase (PI3K) signaling, although it impaired InlB-mediated actin polymerization [45,46]. The requirement of lipid rafts for InlA-mediated entry was further established by the recruitment of caveolin to the entry sites, which was necessary for bacterial internalization. InlB is essential for InlA-dependent entry into cells lacking constitutive PI3-K activity (placenta), while it is not required in cells with activated PI3-K (intestine) [46,47]. Overall, it can be concluded that InlA is an adhesion molecule within cholesterol-rich lipid rafts, triggering *L. monocytogenes* internalization through PI3-K activation and caveolin-mediated endocytosis [48].

*L. monocytogenes* initially invades the host cell by exploiting clathrin-mediated endocytosis, where it initially resides within a vacuole [44]. Subsequently, the bacterium secretes the pore-forming toxins, LLO, which enable it to escape the vacuole and establish itself in the host cell cytoplasm. Once in the cytoplasm, the bacterium spreads to neighboring cells by generating actin-rich protrusions at one of its poles, facilitating its movement from one cell to another [49]. At this stage of infection, the bacterium also employs caveolae to facilitate intercellular movement from one cell to another [49,50]. In non-pagocytic cells, when actin filament-rich protrusions containing the bacteria extend from one cell, they engage with ubiquitinated E-cadherin on adjacent cells. Caveolae play a crucial role by forming flattened invaginations that wrap around these bacterial protrusions, effectively engulfing them (Figure 2). The core proteins of caveolae, such as Cav-1, Cav-2, and subset of the caveolin-associated proteins (cavin-2 and EHD2) are integral to this process, while additional clathrin-interacting proteins like Epsin-1 assist in bending the membrane to create these invaginations. This mechanism significantly enhances the bacterium’s ability to spread from one cell to another, as the actin filament-driven protrusions push the bacteria toward the host cell, deforming its surface [50] (Figure 2).

Unlike the E-cadherin-mediated uptake in non-phagocytic cells, in phagocytic cells, *L. monocytogenes* secrete LLO that damages the plasma membrane of the infected cells and induces PtdSer inversion in the lipid rafts (Figure 2). The bacterial protrusions with PtdSer bind to TIM4, which serves as the PtdSer receptor on the host cell membrane and initiates caveolin-mediated endocytosis to internalize the bacterium [48]. More recently, it was shown that the formin mDia1 is necessary for recruiting Filamin A, an F-actin cross-linking protein associated with caveolae, as well as Cav-1 to membrane invaginations [49].

### 2.2. Mycobacterium tuberculosis

*M. tuberculosis* is the causative agent of tuberculosis, a global threat affecting one-quarter of the world’s population. It utilizes cholesterol-dependent raft microdomains to invade and proliferate within phagocytic and non-phagocytic cells, such as epithelial cells, mast cells, macrophages, dendritic cells, fibroblasts, type-II pneumocytes, and endothelial cells. Once internalized, the bacteria remain localized within caveolae, effectively circumventing the lytic activity of lysosomal enzymes and avoiding destruction by reactive oxygen and nitrogen species [51]. To invade myeloid-derived suppressor cells, *M. tuberculosis* interacts with a receptor embedded in lipid rafts on the host cell surface. Caveolae and associated proteins bind to the intracellular face of the lipid raft through cholesterol-binding interactions. This binding induces spontaneous invagination of the host cell’s plasma membrane, activating dynamin at the neck of the invagination site, which facilitates the formation of vesicles that bud off into the cytosol. This mechanism allows *M. tuberculosis* to evade lysosomal fusion, promoting its survival within the host and enabling it to hijack the host’s immune defenses [52].

Due to genetic similarity to *M. tuberculosis*, *Mycobacterium bovis Bacillus Calmette-Guérin* (BCG) is commonly used as a model organism for studying tuberculosis infection. Recent studies in mice show that Cav-1 regulates apoptosis and the inflammatory response in macrophages infected with BCG. Macrophages in Cav-1-deficient mice exhibit increased bacterial loads in the liver, suggesting that Cav-1 is essential for early elimination and protection against BCG. This process involves Cav-1’s regulation of acid sphingomyelinase (Asm)-dependent ceramide formation, apoptosis, and inflammatory cytokine production following BCG infection [53].

### 2.3. Staphylococcus aureus

*S. aureus* invades lung epithelial cells through lipid raft-mediated endocytosis, enabling it to evade immune defenses. A recent study identified α5β1 integrin and fibronectin-binding proteins (FnBPs) as initial adhesion points, with lipid rafts facilitating bacterial entry [54]. The disruption of lipid rafts reduced *S. aureus* internalization, and colocalization with the lipid raft marker ganglioside GM1 and Cav-1 confirmed this entry route. Alpha-hemolysin (Hla) was crucial for internalization, as Hla-deficient strains showed reduced internalization, implicating Hla’s interaction with Cav-1 in the internalization process [54].

### 2.4. Streptococcus Species

*Streptococci* are a diverse group of Gram-positive pathogens classified into various groups based on their hemolytic properties and serological characteristics. The most medically relevant pathogenic species in humans are grouped into Group A, B, and other non-grouped streptococci. *Streptococcus pyogenes* (Group A *Streptococcus*) causes infections like strep throat, scarlet fever, and more severe conditions like necrotizing fasciitis and toxic shock syndrome. Most of the species of *Streptococcus* utilize caveolae to invade host cells. In Group A *Streptococcus*, the M protein and streptococcal fibronectin-binding protein I (SfbI) are the key virulence factors evading antibiotics and host immune defenses. Sfbl is pivotal in *S. pyogenes* invasion through a caveolae-dependent pathway [55]. SfbI, located on the bacterial surface, interacts with host fibronectin and α5β1 integrins, triggering the clustering of these integrins on the host cell surface [56]. This event promotes the aggregation of caveolae, leading to the formation of specialized membrane-bound compartments known as “caveosomes.” The recruitment and invagination of caveolae around the bacterium facilitates its internalization into the host cell. Disrupting lipid rafts and cholesterol with MβCD and filipin prevented the invasion of *S. pyogenes* into HEp-2 cells [55]. Crucially, this process enables the bacteria to bypass lysosomal fusion, supporting *S. pyogenes* survival within epithelial and endothelial cells [55,56]. A recent study has revealed that the GAS effector protein NAD-glycohydrolase (Nga) acts as a negative regulator in Cav-1-mediated internalization of GAS in human epithelial cells [57].

*Streptococcus agalactiae* (Group B *Streptococcus* (GBS)) commonly colonizes the gastrointestinal and genitourinary tracts in humans. However, in newborns with compromised immune systems, it is a major cause of neonatal infections. GBS predominantly invade polarized cells from their lateral surfaces using α3β1 and α2β1 integrins, which trigger a cellular response involving tyrosine kinase-dependent endocytosis and extensive actin remodeling [58]. Recent findings demonstrate that intact lipid rafts, i.e., flotillin-1 and Cav-1, and the PI3K/AKT pathway are crucial for *S. agalactiae* to invade endothelial cells [59]. Disrupting cholesterol by pre-treating cells with MβCD or inhibiting the PI3K/AKT signaling pathway with LY294002 blocked bacterial invasion [59].

*Streptococcus pneumoniae* (non-grouped streptococci) causes invasive infections such as otitis media, sinusitis, lobar pneumonia, bacteremia, and meningitis. *S. pneumoniae*, invasion, and intracellular trafficking within respiratory epithelial cells involve caveolin-mediated and clathrin-mediated endocytosis (CME). The pneumococcal surface protein C (PspC) plays a crucial role in interacting with the polymeric immunoglobulin receptor (pIgR) on the host cell surface, facilitating bacterial invasion [60]. *S. pneumoniae* uptake via the PspC–pIgR pathway involves active dynamin-dependent caveolae and clathrin-coated vesicles. Cholesterol depletion from host cell membranes and the disruption of lipid microdomains hindered pneumococcal internalization. Additionally, chemical inhibition of clathrin, or the functional inactivation of dynamin, caveolae, or clathrin via RNA interference (RNAi), significantly reduced pneumococcal uptake, indicating the involvement of both CME and caveolae in the invasion process [60].

## 3. Gram-Negative Bacterial Pathogens Manipulate Caveolin-Mediated Endocytosis to Infiltrate Host Cells, Secure a Niche for Survival, and Evade the Immune System

### 3.1. Brucella *spp*.

*Brucella suis* is a Gram-negative facultative intracellular bacterium that survives and replicates within a membrane-bound compartment inside professional and nonprofessional phagocytic cells. By using cholesterol-sequestering drugs (filipin and- MβCD) and GM1binding (cholera toxin B) molecules and manipulating the lipid raft components, i.e., cholesterol and ganglioside GM1, it was found that lipid rafts, specifically under non-opsonic conditions, serve as entry points that allow the bacteria to prevent phagosome–lysosome fusion [61]. This suggests lipid rafts promote a stable environment within the host cell by restricting phagosome maturation at the membrane, supporting the short-term survival of *B. suis*. Moreover, lipid raft components like GPI-anchored proteins, GM1 gangliosides, and cholesterol are incorporated into these macropinosomes, while proteins like LAMP-1 and CD44 were excluded [62]. Disrupting lipid raft elements reduced VirB-dependent macropinocytosis and replication, implicating lipid raft-mediated entry in the pathogen’s intracellular survival [62]. In contrast, the phagocytic trafficking and the intracellular survival of *Brucella abortus* are primarily mediated by clathrin and Rab5-mediated-mediated endocytosis [63].

### 3.2. Campylobacter jejuni

*C. jejuni* is a Gram-negative foodborne pathogen that causes human diarrheal diseases and is the leading cause of Guillain–Barre’s paralysis. Caveolae plays an essential role in the 81–176 strain of *C. jejuni* internalization in intestinal epithelial cells by serving as a special microdomain and facilitating the bacterium’s interaction with the host membrane receptor located within caveolae. The disruption of membrane caveolae via the pretreatment of intestinal cell monolayers with filipin III reduced pathogen entry by 95%, suggesting that caveolae-mediated interaction triggers signal transduction events that lead to cytoskeleton rearrangements, which are necessary for bacterial internalization [64]. Moreover, after internalizing epithelial cells via a caveolae-dependent pathway, *C. jejun*i-containing vacuole avoids lysosomal degradation and resides in a unique intracellular pathway that deviates from the canonical endocytic pathway [65]. Conversely, in macrophages, the bacteria are sent to lysosomes and swiftly eliminated, suggesting evolved strategies for survival in various host cell environments [65].

### 3.3. Chlamydia trachomatis

*C. trachomatis* is an obligate intracellular bacterial pathogen and a major causative agent of sexually transmitted infections [66]. The involvement of caveolae in the entry of *C. trachomatis* is an ongoing debate. Prior studies demonstrated that *C. trachomatis* serovar K invades the non-phagocytic human epithelial cells (HEp-2 and HeLa 229) and the phagocytic mouse macrophage cells (J-774A.1) by using cholesterol- and sphingolipid-rich lipid rafts containing caveolin [67]. Drugs that disrupt these lipid rafts by depleting cholesterol hindered chlamydial entry. The bacteria then form vesicles marked by caveolin, bypassing lysosomal fusion and acidification. Instead, they travel to the Golgi region, acquiring lipids and cholesterol. They maintain high levels of caveolin, typically recycled back to the plasma membrane, which supports their survival and replication.

Consistent with the above observations, another study reported that in HeLa cells infected with different *Chlamydia* species, both Cav-1 and Cav-2 are found at the inclusion membranes of *C. pneumoniae*, *C. caviae*, and *C. trachomatis* serovars E, F, and K [68]. Only Cav-2 colocalized with *C. trachomatis* serovars A, B, and C. This suggests a specific or indirect role for Cav-2 associated with these pathogens, possibly supporting their survival or replication at the inclusion membranes within host cells. However, another study using RNA interference (RNAi), immunoblotting, immunofluorescence, and RT-PCR showed that key structural elements of the clathrin-mediated endocytic pathway, such as clathrin heavy chain, dynamin-2, heat shock 70-kDa protein 8, Arp2, and cortactin, but not caveolae, are not involved in *C. trachomatis’* entry into the host cells [69].

### 3.4. Edwardsiella tarda

*E. tarda* is a Gram-negative bacterium known to cause bacteremia. As an intracellular pathogen, it can invade phagocytic and non-phagocytic cells, replicating within host cells. *E. tarda* primarily utilizes clathrin-mediated endocytosis for invasion, though it has also been shown to use caveolin-mediated endocytosis as an alternative, mainly when clathrin-dependent pathways are limited [70]. The invasion of *E. tarda* into macrophages was significantly reduced when clathrin- and caveolin-mediated endocytic pathways and endosome acidification were inhibited. However, blocking macropinocytosis did not affect bacterial invasion. A related species, *Edwardsiella piscicida*, a common fish pathogen, relies on caveolin-mediated pathways to enter non-phagocytic cells. Inhibitory drugs and shRNA-mediated downregulation to block specific endocytic pathways showed that the bacterium enters non-phagocytic cells through micropinocytosis and caveolin-mediated endocytosis, requiring cholesterol and dynamin for successful invasion [71].

### 3.5. Ehrlichia caffeensis and Anaplasma phagocytophilum

*E. caffeensis* and *A. phagocytophilum* are obligatory intracellular bacteria that are agents of human monocytic ehrlichiosis (HME) and human granulocytic ehrlichiosis (HGE), respectively [72]. These emerging tickborne zoonoses cause fever, headache, anorexia, and chills and are frequently accompanied by anemia and elevations in serum hepatic aminotransferases. To enter the host cell, *E. caffeensis* and *A. phagocytophilum* bind to their specific receptor located in the host cell surface caveolae that trigger activation of downstream signaling and enter the host cell by caveolae-mediated endocytosis that does not fuse with lysosomes [73]. Using fluorescence microscopy, Cav-1 was shown to co-localize with both early-stage and replicative bacterial inclusions. Additionally, proteins phosphorylated by tyrosine and PLC-γ2 were detected within the early inclusions. Clathrin was absent from all inclusions, suggesting a non-clathrin-mediated entry mechanism.

Furthermore, bacterial proteins from both *E. chaffeensis* and *A. phagocytophilum* were found to co-fractionate with Triton X-100-insoluble raft fractions. This suggests that caveolae encapsulating the bacteria retain essential signaling molecules required for entry and facilitate interactions with the recycling endosome pathway, which may contribute to the survival of these obligate intracellular pathogens within the host’s immune response [73]. Recently, using yeast two-hybrid screening, co-immunoprecipitation, antibody blocking, and enzymatic inhibition, it was demonstrated that caveolae-dependent uptake of *A. phagocytophilum* via the invasion AipA9–21 and its interaction with host CD13 induces Src kinase signaling that mediates uptake into host cells [74].

### 3.6. Escherichia coli

Uropathogenic *Escherichia coli* (UPEC) is associated with a wide range of infections in humans beyond urinary tract infections (UTIs) that include myositis, skin structure infection, osteomyelitis, epididymal orchitis, and meningitis. Severe *E. coli* infections lead to systemic inflammatory response syndrome (SIRS), which causes significant mortality [75]. UPECs that express type 1 pili are the primary causative agent of UTIs. To attach and invade host urinary bladder epithelial cells at the distal tip of each type 1 pilus, an adhesion protein known as FimH (fimbrial adhesin H) interacts with mannosylated glycoproteins, triggering bacterial entry [76].

CD48 is a GPI-anchored protein localized in plasmalemma caveolae in host cells, facilitating binding in FimH-expressing *E. coli* [77]. It forms a large bacteria-encapsulating compartment by recruiting intracellular vesicular caveolae to the bacterial attachment site. The involvement of caveolae in bacterial uptake was confirmed by immunoelectron microscopy, and several caveolin-specific markers, such as caveolin, GM1, and cholesterol, were observed to mobilize and encapsulate the bacteria. Cell fractionation studies have suggested that the disruption of caveolae through cholesterol depletion inhibits bacterial uptake, confirming their involvement in the endocytic process that avoids classical lysosomal degradation pathways, thereby allowing the bacteria to remain viable intracellularly [28,77].

*Escherichia coli* K1 strain can cause meningitis by crossing the human blood–brain barrier (BBB). In the human brain, microvascular endothelial cells (HBMEC) that form the BBB, *E. coli* K1, bind to Gp96-like receptors, which cluster within caveolae. The bacterium triggers the activation of PKCα, which interacts with Cav-1. Overexpression of a dominant-negative version of Cav-1 with mutations in its scaffolding domain inhibited the interaction between phospho-PKCα and Cav-1 and prevented *E. coli* from invading HBMEC [78]. This interaction helps reorganize the actin cytoskeleton at the bacterial entry site and enwrap the bacteria that form caveolae. When bacteria internalize into caveolae, they avoid lysosomal fusion, enabling transcytosis across the BBB without being degraded, which aids their survival and spread within the host [78].

In addition to the Gp96-like receptor, the interaction of the virulence factor, namely, invasion of brain endothelial cells A (IbeA), and its primary receptor vimentin is the upstream signaling event that is required for the caveolae/lipid raft (LR)-dependent entry of *E. coli* K1 into HBMECs [79]. Vimentin is within a unique lipid raft containing cholesterol, sphingolipids, and Cav-1. After binding of IbeA with vimentin, caveolae on the host cell membrane are actively involved in clustering around the binding site and initiate a downstream signaling cascade, finally recruiting other proteins such as Cav-1, a7 nicotinic acetylcholine receptor (a7 nAChR), and polypyrimidine tract-binding protein-associated factor (PSF) into the lipid raft. Subsequently, the NF-κB signaling pathway is activated through TAK1 and ERK. This process enhances bacterial penetration and leucocyte transmigration across the BBB and causes subsequent inflammation [80].

In *E. coli* K12-infected monocytes, the pattern-recognition receptor, Toll-like receptor 4 (TLR4), enhances caveolae-mediated endocytosis and bacterial uptake. Blocking TLR4, Src signaling, or the caveolae-mediated endocytosis pathway in transgenic sheep monocytes results in reduced bacterial uptake, a diminished capacity to clear bacteria, and increased endosomal pH. This indicates that caveolae-mediated endocytosis is crucial for effective bacterial internalization, as well as the destruction and maintenance of an acidic endosomal environment required for the antimicrobial activity of monocytes [81].

### 3.7. Francisella tularensis

*F. tularensis* is a Gram-negative, highly infectious bacterium that causes tularemia. *Francisella* relies on Cav-1 during its invasion of phagocytic cells but is dispensable in non-phagocytic cells [82]. Consistent with these observations, Tamilselvam et al. show that in murine macrophages during entry, and in the early stages of intracellular transport within the host cell, components associated with lipid rafts, such as cholesterol and Cav-1, were incorporated into vesicles containing *Francisella* [83]. Interfering with lipid rafts by depleting plasma membrane cholesterol, inducing raft internalization with cholera toxin, or removing raft-associated GPI-anchored proteins using phosphatidylinositol phospholipase C significantly reduced the entry of *Francisella* and its ability to multiply intracellularly [83]. After entering the host cell, *F. tularensis* resides within the phagosome. Shortly after uptake, it disrupts the phagosomal membrane and escapes into the cytoplasm, where it replicates [84].

### 3.8. Helicobacter pylori

*Helicobacter pylori* is a Gram-negative bacterium widely recognized for its role in chronic active gastritis, which can progress to chronic atrophic gastritis and intestinal metaplasia with prolonged infection [60]. Membrane vesicles have been observed in gastric biopsy specimens from infected individuals, highlighting the importance of the endocytic pathway in the bacterium’s invasion of host cells. Detailed investigations into the mechanisms of endocytosis implicated in *H. pylori* invasion have identified both clathrin-dependent and clathrin-independent pathways as contributors to the uptake of *H. pylori* vesicles. Notably, knockdown experiments in human gastric adenocarcinoma cell line AGS have demonstrated that clathrin-mediated endocytosis is predominant in vesicle internalization [85]. Moreover, confocal microscopy showed that *H. pylori*-derived vesicles exhibited colocalization with clathrin and dynamin-II and markers corresponding to the subsequent stages of endosomal and lysosomal trafficking [85]. However, colocalization of Cav-1 with *H. pylori* vesicles was also observed, suggesting that caveolin-mediated endocytosis may also be involved in this process [85].

### 3.9. Klebsiella pneumonia

*K. pneumoniae* is the third most commonly isolated bacterium in the blood of patients with sepsis. Using fluorescent microscopy, Huang Weaver et al. demonstrated that Cav-1 plays a role in the internalization of the pathogen into alveolar epithelial cells [85]. Although the transfection of dominant-negative Cav-1 significantly reduced bacterial internalization into lung cells, Cav-1 in the lipid raft also influenced DNA damage and repair responses by regulating reactive oxygen species, cell death, and inflammatory reactions, which is critical to host defense against *K. pneumonia* [85]. Along similar lines, recent studies using Cav-1 knockout mice showed that these mice exhibited significantly poorer outcomes following *K. pneumoniae* infections, with reduced survival, increased bacterial load, exacerbated tissue damage, heightened proinflammatory cytokine responses, and extensive systemic bacterial dissemination [86].

### 3.10. Leptospira

*Leptospira* is an emerging Gram-negative bacterium responsible for leptospirosis, a zoonotic disease that can spread through the bloodstream to organs such as the lungs, liver, kidneys, and even cerebrospinal fluid, exacerbating the illness [87]. To invade human and mouse blood vessel endothelial cells, renal tubule epithelial cells, and fibroblasts, *Leptospira* species employ a caveolae/integrin-β1-PI3K/FAK-microfilament endocytosis pathway (Figure 3). Once internalized, *Leptospira* forms specialized vesicles to avoid lysosome fusion; these vesicles utilize Rab5/Rab11 and Sec/Exo-SNARE proteins to assist in recycling and transport within the cell. Eventually, *Leptospira* exits the host cell through a FAK/microfilament/microtubule pathway mediated by the SNARE complex [87] (Figure 3). When entering vascular endothelial cells, *Leptospira* interacts with integrins and Cav-1 on the cell surface, promoting its internalization. In human umbilical vein endothelial cells (HUVECs), integrin β1 and Cav-1 aid bacterial entry by activating PI3K signaling, leading to cytoskeletal rearrangement through actin polymerization, which is an essential process for bacterial internalization and transcytosis [88].

### 3.11. Neisseria gonorrhea

*N. gonorrhea* is a Gram-negative obligate human pathogen that causes the sexually transmitted infection gonorrhea. The bacterium invades host cells through a complex mechanism that involves caveolae. The PorBIA outer membrane protein on the bacterial surface binds to the host scavenger receptor SREC-I, leading to the recruitment of Cav-1 [90]. The presence of pili modulates the role of Cav-1 in bacterial invasion. When pili are absent, Cav-1 becomes phosphorylated and recruits PI3-K to facilitate the downstream signaling required for bacterial uptake that promotes bacterial uptake by activating protein kinase D1 (PKD1). This uptake process is regulated by signaling pathways involving phospholipase C gamma 1 (PLCγ1), PI3-K, and PKD1, which work together to remodel the actin cytoskeleton, facilitating *N. gonorrhea* invasion [91]. Cav-1 also plays a role in the formation of ceramide-enriched membrane domains that compartmentalize receptors and signaling molecules, which is necessary for PorBIA-mediated bacterial invasion by concentrating SREC-I and organizing the signaling components involved in bacterial invasion [91].

### 3.12. Porphyromonas gingivalis

*P. gingivalis* is a bacterial pathogen responsible for human chronic periodontal disease. This pathogen employs multiple pathways to invade host cells, where caveolae play a crucial role. The bacterial fimbriae bind to ICAM-1 to invade host oral epithelium cells. Through this interaction, Cav-1, within caveolae vesicles, internalizes *P. gingivalis*, enabling the bacterium to progress to early endosomes and eventually reach autophagosomes (Figure 3). The attachment of fimbriae also activates TLR2, which subsequently increases ICAM-1 expression and promotes clustering on the host cell surface [92].

Recent elaborate studies by Lei et al. demonstrated that *P. gingivalis* infected brain microvascular endothelial cells (BMECs) in vitro and in vivo showed increased caveolae and higher Cav-1 expression [89]. Downregulation of Cav-1 levels reduced *P. gingivalis*-mediated BBB permeability. *P. gingivalis* arginine-specific gingipain (RgpA) colocalized with Cav-1. Additionally, *P. gingivalis* significantly lowered the expression of an essential host protein, the major facilitator superfamily domain containing 2a (Mfsd2a)m which is critical for maintaining BBB integrity (Figure 3). Overexpression of Mfsd2a reduced both Cav-1 expression and *P. gingivalis*-mediated BBB permeability. The authors suggested that the Mfsd2a/Cav-1 transcytosis pathway is central to *P. gingivalis*-induced BBB permeability and pathogen entry, leading to neurological damage [89].

### 3.13. Pseudomonas aeruginosa

*Pseudomonas* is a Gram-negative bacterium that mainly causes pneumonia. The pathogen enters the host cells via lipid raft-mediated endocytosis. The activated Cav-2 forms a lipid raft on the host cell membrane that promotes *P. aeruginosa* entry, allowing bacteria to evade host immune defense and replicate intracellularly [93]. In epithelial cell internalization, Cav-1 facilitates the formation of an internalization platform with the cystic fibrosis transmembrane conductance regulator (CFTR) that promotes endocytosis [94].

*P. aeruginosa* employs lipid rafts, primarily intact lipid raft platforms and Cav-2, to invade type I pneumocytes in the lungs. Phosphorylation of Cav-2 is a critical step for regulating lipid raft-mediated endocytosis of bacterium. This mechanism assists *P. aeruginosa* invasion of the alveolar cells, leading to severe conditions like pneumonia and sepsis. When *P. aeruginosa* enters these cells, it colonizes with lipid raft components, which protect the bacteria from being cleared by immune cells [25].

Through MS analysis, Thuenauer et al. recently demonstrated that LecB on the bacterial cells binds multiple apical receptors on the host cells (such as CEACAM1, MUCIN-1, ICAM1, and podocalyxin). This receptor clustering triggers the Src-PI3K-Rac signaling cascade, which leads to actin rearrangement and membrane protrusion formation that promotes bacterial uptake. On the apical plasma membrane, LecB also recruits Cav-1 to the apical plasma membrane, which is also critical in PI3-K activation and further facilitates endocytosis and bacterial uptake [95]. However, in vivo studies with *Cav-1* knockout mice revealed that the absence of Cav-1 in the lungs and spleen increased sensitivity to *P. aeruginosa* infection, increased mortality rate, elevated bacterial burdens, and elevated inflammatory responses via the Cav-1/STAT3/NF-κB axis [96,97].

### 3.14. Rickettsia *spp*.

*Rickettsia* species are Gram-negative obligate intracellular bacteria, with *R. conorii* and *R. rickettsii* being the primary species responsible for the Mediterranean and rocky mountain spotted fever, respectively [98]. These bacteria primarily infect the microvascular endothelium that lines blood vessels. They enter host cells using fibroblast growth factor receptor 1 (FGFR1)- and Cav-1-mediated endocytosis. Sahni et al. demonstrated that pathogenic spotted fever group (SFG) rickettsiae can interact with heparan sulfate proteoglycans (HSPG) and the FGFR1 complex on the host cell surface, facilitating their internalization through FGFR1/Cav-1-mediated endocytosis [99]. Proteomic analysis revealed that the β-peptide of the rickettsial outer membrane protein A (OmpA) interacts with FGFR1, aiding in host cell invasion. Additionally, independent silencing of Cav-1 and Cav-2 suggests that rickettsiae may also use the Ku70 receptor for entry via Cav-2-dependent endocytosis [99]. Martinez et al. further demonstrated that inhibiting Ku70 impairs the internalization of *R. conorii* into host cells, indicating the involvement of Ku70-associated Cav-2-mediated endocytosis [100].

### 3.15. Salmonella enterica serovar Typhimurium

The Gram-negative facultative intracellular bacterium belonging to the non-typhoidal *Salmonella* family is a significant cause of foodborne illnesses globally. In the US alone, it causes as many as 1.35 million infections and 420 deaths annually [30]. The pathogen primarily spreads through contaminated food sources like poultry, eggs, and produce, leading to symptoms like diarrhea, fever, and abdominal cramps within 6 to 48 h of infection.

*S. enterica* invade nonphagocytic cells by delivering the effector proteins SopE, SopE2, and SopB into the host cell through a type III secretion system (TTSS), encoded by pathogenicity island I (SPI-I). These effectors activate signaling pathways within the host cell, leading to various responses, notably the formation of membrane ruffles rich in actin that enable bacterial entry via a “trigger”-like mechanism. At the bacterial attachment site, caveolae are present in the host cell membrane as an invagination cluster at the site of interaction and help recruit other core proteins like Cav-1 and Rac1 to the interaction site [101]. Bacterial effector protein SopE, with these core proteins, work together to rearrange the actin cytoskeleton, forming a membrane ruffle that allows bacteria to be engulfed by the host cells. However, unlike *E. coli*, *S. enterica* only manipulates caveolae to attach and invade the host cell; they do not use caveolae to survive inside the host cell. Once *S. enterica* is fully internalized, SopE levels decline in the caveolae and host cell cytoplasm, accompanied by a decrease in Rac1 activity. Reducing Cav-1 expression through siRNA treatment led to a lower *Salmonella* invasion rate than cells treated with control siRNA.

While *S. enterica* serovar Typhimurium utilizes Cav-1 to enter non-phagocytic host cells through the apical plasma or basolateral membrane, Cav-2 expressed on the basolateral membrane inhibits Cdc42 activation and decreases pathogen uptake as Cdc42 promotes bacterial uptake and membrane ruffling. Knockdown of Cav-2 retards intestinal epithelial cell proliferation and increases bacterial uptake [102]. Interestingly, to counteract the decreased Cav-2 expression, the bacterium induces miR-29a transcription, relieving Cdc24 inhibition and increasing bacterial invasion [102,103].

In addition to classical enterocytes, *S. enterica* serovar Typhimurium crosses the intestinal barrier through the M-cells of Peyer’s patches, the specialized immune cells in the epithelium covering Peyer’s patches of the small intestine. siRNA-mediated downregulation of Cav-1 in an in vitro model of the M-cell-like model showed a significant reduction in bacterial transcytosis and uptake [104]. Moreover, in aged mice, Cav-1 showed high expression levels in the Peyer’s patches and spleen, and disruption of Cav-1 either by MβCD or siRNA in senescent non-phagocytic cells showed a marked reduction of bacterial invasion [104]. These findings imply that elevated levels of Cav-1 in aging cells may contribute to the susceptibility of aged populations to microbial infections.

Interestingly, mice lacking Cav-1 had significantly higher bacterial loads in their spleen and other tissues when infected with *S. enterica* serovar Typhimurium, compared to WT mice when challenged intravenously [105]. Although Cav-1-deficient mice had increased production of inflammatory signals, including chemokines and nitric oxide, they still suffered from a more severe infection and lower survival rates. In these mice, neutrophil infiltration into granulomas and liver damage increased, as indicated by higher levels of necrosis. Cav-1-deficient macrophages showed an exaggerated inflammatory response and produced more nitric oxide in response to bacterial LPS in laboratory conditions. These findings suggest that Cav-1 plays a crucial role in controlling the immune response, particularly in macrophages, and that the lack of Cav-1 leads to an overproduction of inflammatory mediators. This excessive production of toxic mediators from macrophages lacking Cav-1 contributes to the heightened susceptibility of Cav-1-deficient mice to *S. enterica* serovar Typhimurium [105].

### 3.16. Shigella flexneri

*S. flexneri* causes bacillary dysentery and, like *L. monocytogenes*, manipulates the host’s actin cytoskeleton, forming actin-rich “comet tails” for intracellular movement and then creating actin-based protrusions at the plasma membrane to move directly between neighboring host cells using caveolae. When a bacterial membrane protrusion encounters the membrane of the host cell, caveolin aids in physically distorting the plasma membrane around the bacterium providing a source of the membrane to release into the large invagination [106]. Like *L. monocytogenes* in *S. flexneri* endocytic invaginations, Cav-1 is colocalized along with other proteins and lipids, such as cavin-2, EHD2, dynamin-2, epsin-1, PtdSer, and PtdIns, as well as actin and the actin-associated proteins VASP and α-actinin-1, excluding α-actinin-4, which is only observed in the case of *L. monocytogenes* [106]. Moreover, in contrast to *L. monocytogenes*, caveolin, cavin-2, and EHD2 localize to the clathrin-rich membrane invaginations formed during *S. flexneri* infections, suggesting a co-occurrence of clathrin and caveolin pathways at the single-endocytic site and a rare case of these typically distinct pathways collaborating. The combined presence of both pathways may enhance structural stability, cargo specificity, or adaptability in response to infection.

## 4. *Mycoplasma* spp.; The Cell-Wall-Less Bacterial Pathogens That Capitalize on Caveolin-Mediated Endocytosis for Cellular Entry

### Mycoplasma *spp*.

Mycoplasmas are among the smallest self-replicating bacteria, associated with a range of diseases, including acute respiratory illnesses, joint infections, and genital and urinary tract infections. Lacking a cell wall, they are classified within the mollicutes class [107]. Mycoplasmas act as surface parasites and exhibit pathogenicity by invading various host cell types, enabling them to survive as intracellular pathogens. Following adhesion to host cell receptors, mycoplasmas stimulate cellular cascade events involving microfilaments, microtubules, and kinases. They primarily utilize clathrin- and caveolin-mediated endocytosis pathways to enter host cells, with vesicles subsequently transported to early endosomes. After internalization, mycoplasmas reside within vesicles, with some being released into the extracellular environment via exocytosis after fusing with recycling endosomes, while others progress to late endosomes and fuse with lysosomes to sustain survival within host cells. Ultimately, mycoplasmas thrive, replicate intracellularly, and may localize within the nucleus [108].

Raymond et al. highlighted the critical role of integrin β1-fibronectin complexes in the trafficking and endocytosis of *Mycoplasma hyopneumoniae* [109]. To invade porcine kidney epithelial cells (PK-15), *M. hyopneumoniae* employs fibronectin-binding proteins to interact with integrin β1 on PK-15 cells. This interaction facilitates integrin clustering, intracellular signaling, and bacterial uptake via clathrin- or caveolin-mediated endocytosis. Their findings demonstrate the co-localization of integrin β1 with fibronectin in PK-15 cells infected with *M. hyopneumoniae*, where the bacteria assemble extracellularly. Blocking integrin β1 in PK-15 cells effectively prevents *M. hyopneumoniae* entry, confirming the essential role of the integrin β1-fibronectin interaction during host cell invasion [109].

## 5. Conclusions and Future Directions

Caveolae are small invaginations on the cell surface primarily formed by caveolins, which serve as the core structural components. Caveolin synthesis begins in the endoplasmic reticulum, followed by oligomerization in the Golgi apparatus. Once oligomers bind to cavin-1, caveolae are formed at the plasma membrane, where cholesterol and phospholipids are essential in assembling distinct lipid pools. Caveolae have multiple functions, including roles in endocytosis, cholesterol and lipid metabolism, mechanosensation, and cellular signaling. Among these, caveolae-mediated endocytosis has garnered significant attention due to its utilization by various microorganisms to invade host cells and evade immune responses (Table 1). In this review, we explored the different roles of caveolae in microbial invasion and immune system evasion.

Among Gram-positive organisms, *L. monocytogenes* employ caveolae to invade non-phagocytic cells, cross the intestinal barrier, and spread cell-to-cell (Table 1). Similarly, *M. tuberculosis*, GAS, and *S. pneumoniae* use caveolae for invasion and survival within host cells after entry (Table 1). Different Gram-negative bacteria use slightly different mechanisms to enter host cells (Table 1). Among all the bacterial entry mechanisms, *S. enterica Serovar Typhimurium* demonstrates a unique and intriguing entry mechanism into host cells compared to other Gram-negative pathogens. In this case, Cav-1 and Cav-2 play opposing roles in bacterial entry into host cells. *S. enterica serovar Typhimurium* utilizes Cav-1 to facilitate its entry through the apical or basolateral plasma membrane. In contrast, Cav-2, located on the basolateral membrane, inhibits bacterial uptake by suppressing the activation of Cdc42. Thus, Cav-1 and Cav-2 function antagonistically in regulating *S. enterica* Serovar Typhimurium entry into host cells. On the other hand, *P. aeruginosa* and *Rickettsia* use both Cav-1 and Cav-2 for internalization into host cells. A few Gram-negative bacterial pathogens, like *K. pneumoniae*, *E. coli* K1, *P. aeruginosa*, and *S. enterica* serovar Typhimurium, have evolved sophisticated mechanisms to hijack caveolin for the modulation of immune response and immune evasion (Table 1).

In addition to these bacterial pathogens, many bacterial toxins exploit the Cav-1-mediated endocytic route for entry [28]. To internalize into the host cell, the cholera toxin (CT) binds the GM1 receptor in caveolar membrane domains on the host cell membrane. This activates adenyl cyclase on the cytosolic surface of the basolateral membrane, leading to an increase in intracellular cAMP, which, in intestinal crypt cells, induces electrogenic Cl^-^ secretion, which leads to the massive secretory diarrhea seen in cholera. Like cholera toxin, aerolysin, the pore-forming toxin produced by *Aeromonas hydrophila*, and VacA toxin, produced by *Helicobacter pylori*, which damages gastric-epithelial cells, exploit the caveolin-dependent pathway to facilitate their entry and modulate its downstream effects on cellular signaling and immune response [28].

Many bacterial pathogens can cross the blood–brain barrier (BBB) and infect the brain, yet the role of caveolae in the invasion or translocation process is still unclear [111]. For example, *Streptococcus*, *E. coli*, and *L. monocytogenes* can all cause meningitis by crossing the BBB. While the role of caveolae in *E. coli* invasion is well established, the involvement of caveolae in *Streptococcus* and *L. monocytogenes* across the BBB is still unclear. This gap in our knowledge presents an exciting opportunity for future research. Recent reports of an unusual co-occurrence of the clathrin and caveolin pathways utilized by *S. flexneri* at a single endocytic site suggest the collaboration of these pathways [106]. Investigating the potential synergistic functions of these pathways may provide new insights into pathogen–host cell interactions and broader endocytic mechanisms.

Given its role in bacterial pathogenesis, Cav-1 has been proposed as a potential target for therapeutic interventions. By targeting the caveolae-mediated entry pathways, it might be possible to block bacterial invasion or disrupt the intracellular niches that protect bacteria from host defenses. For example, inhibitors of caveolae formation or Cav-1 function could prevent certain pathogens from gaining access to host cells. Along similar lines, a recent study showed that understanding the internalization mechanisms through Cav-1-dependent pathways was crucial for optimizing the delivery of the human papillomavirus HPV16E7 affibody that enters cells to target intracellular proteins and neutralizes the HPV16 E7 oncoprotein, potentially stopping the progression of cancer at an early stage [112]. Modulating Cav-1 activity may also alter the immune response to enhance the host’s ability to clear infections. For example, Cav-1 has been suggested as a target for treating acne [113].

Caveolin, particularly Cav-1, is a critical factor in bacterial pathogenesis (Table 1). It provides pathogens with a means to invade host cells, evade immune detection, and establish chronic infections. By manipulating caveolae-mediated pathways, bacterial pathogens invade host cells, create favorable intracellular environments, modulate host signaling processes, and evade immune defenses. Further research into understanding the function of caveolin in bacterial infections deepens our understanding of host–pathogen interactions and opens new avenues for therapeutic strategies to disrupt these processes.

## Figures and Tables

**Figure 1 cells-14-00002-f001:**
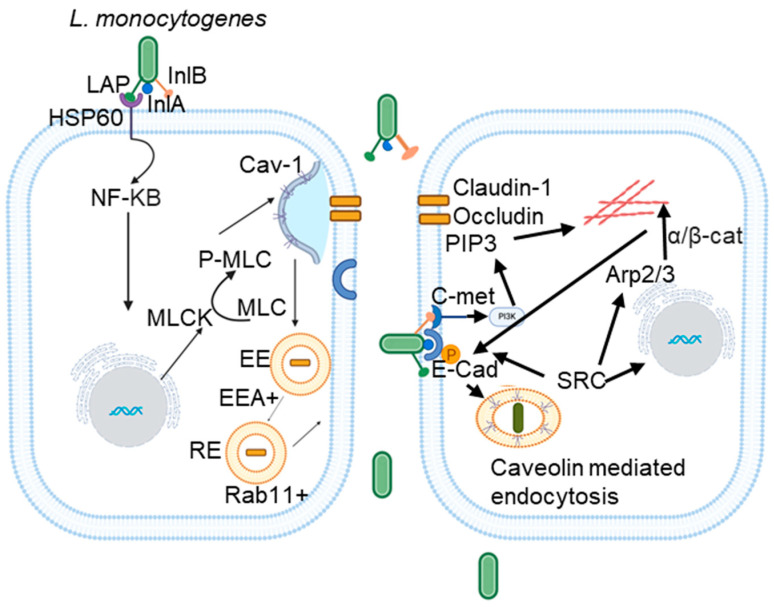
Schematic depicting the mechanism of *L. monocytogenes* LAP and caveolin-mediated translocation across the intestinal epithelial barrier and subsequent InlA-mediated internalization across non-phagocytic cells. LAP on *L. monocytogenes* binds to its host cell surface receptor heat shock protein 60 (Hsp60), inducing endocytosis of tight junction proteins, claudin-1, occludin, and the adherens junction protein E-cadherin via caveolin-1 and MLCK-mediated endocytosis. This disrupts cell junctions, allowing *L. monocytogenes* to pass through the paracellular spaces. InlA subsequently binds to its receptor E-cadherin at the adherens junctions to mediate transcytosis across the epithelial barrier. In non-phagocytic cells, the bacterial surface protein InlA and InlB interact with E-cadherin and c-met, leading to the cytoskeletal rearrangement via a zipper mechanism that triggers *L. monocytogenes* internalization through PI3-K activation and caveolin-mediated endocytosis. Figure created using Biorender and adapted from [43].

**Figure 2 cells-14-00002-f002:**
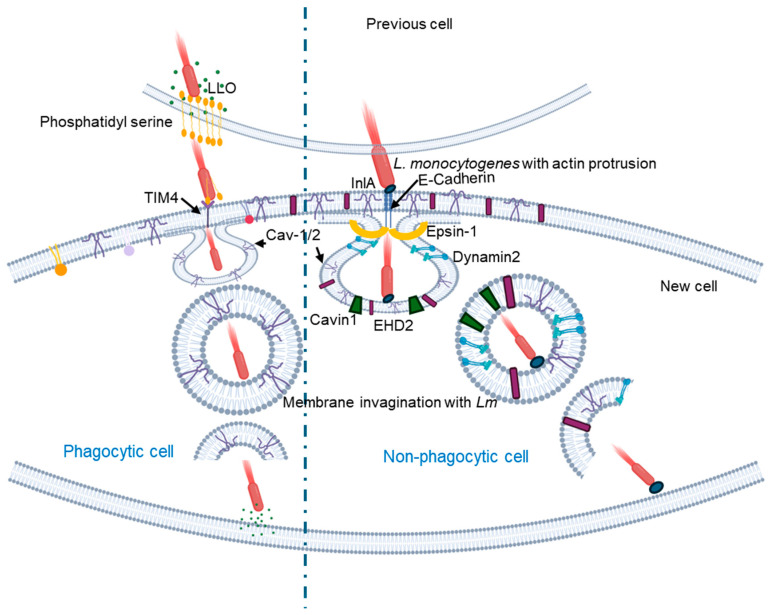
Schematic representation of the cell-to-cell spread mechanism of *L. monocytogenes* in phagocytic and non-phagocytic cells. In phagocytic cells (**left**), internalized actin protrusions containing *L. monocytogenes* secrete LLO, which disrupts phosphatidyl serine on the plasma membrane. Both actin protrusions and phosphatidyl serine-positive *L. monocytogenes* bind to the TIM4 receptor on the host cell surface, which causes internalization of *L. monocytogenes* via caveolin-mediated endocytosis. In non-phagocytic cells (**right**), when actin filament-rich protrusions containing the bacteria extend from one cell, they bind to ubiquitinated E-cadherin in adjacent cells. This binding triggers caveolae to form a flattened invagination that wraps around these bacterial protrusions, effectively engulfing them with the help of some core proteins of caveolae, such as Cav-1, Cav-2, a subset of the caveolin-associated proteins (cavin-2 and EHD2), and clathrin-interacting Epsin that assists in bending the membrane to create these invaginations. Figure created using Biorender.

**Figure 3 cells-14-00002-f003:**
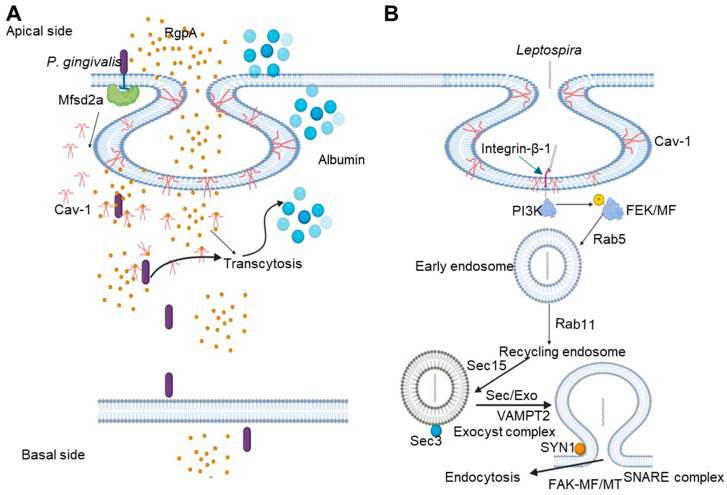
Schematics depicting the internalization mechanism of *P. gingivalis* and *Leptospira* via caveolin-mediated endocytosis. (**A**) The interaction of the virulent factor RgpA of *P. gingivalis* with Cav-1 in the host cell facilitates the internalization of *P. gingivalis* via caveolae. *P. gingivalis* inhibits the integrity of Mfsd2a, leading to enhanced transcytosis across the blood–brain barrier and increased Cav-1 expression, which induces albumin uptake to the cell (adapted from [89]). (**B**) Leptospiral species interacts with integrin-β-1 on host cells; it triggers caveolin to form an invagination; and through the caveolae/integrin-b1-PI3K/FAK-microfilament endocytosis pathway, it enters the host cell. To avoid fusion with lysosomes, it forms leptospiral vesicles inside the host cell, and these vesicles recruit Rab5/Rab11 and Sec/Exo-SNARE proteins in endocytic recycling and vesicular transport systems for intracellular migration and finally release from the cells through a SNARE complex-mediated FAK/microfilament/microtubule endocytosis pathway. Figure created using Biorender.

**Table 1 cells-14-00002-t001:** A comparative table of bacterial pathogens’ exploitation of caveolin-mediated endocytosis for invasion, intracellular survival, cell–cell spread, and the modulation of immune responses.

Pathogen	Pathway	Caveolin Involved	Reference
Bacterial invasion and intracellular survival			
*Anaplasma phagocytophilum*	Bacterial internalization and intracellular survival within caveosome	Cav-1	[73]
*Brucella* spp.	Caveolin-mediated entry	Cav-1	[61,62]
*Campylobacter jejuni*	Helps in bacterial internalization and intracellular survival	Cav-1	[64,65]
*Edwardsiella tarda*	Caveolin-mediated invasion and intracellular survival	Cav-1	[70,71]
*Ehrlichia chaffeensis*	Bacterial internalization and intracellular survival within caveosome	Cav-1	[73]
*Escherichia coli*	Caveolin-mediated invasion and intracellular survival	Cav-1	[78,80]
*Fransicella tularensis*	Caveolin-mediated entry into macrophages and hepatocytes; proliferation inside macrophages	Cav-1	[82,83]
*Helicobacter pylori*	Caveolin-mediated entry into human gastric adenocarcinoma cell line	Cav-1	[110]
*Klebsiella pneumonae*	Caveolin-mediated internalization	Cav-1	[85]
*Leptospira*	Caveolin-mediated entry	Cav-1	[88]
*Listeria monocytogenes*	Apical junctional remodeling for bacterial translocation and internalization.	Cav-1	[43,45]
*Mycoplasma* spp.	Caveolin-mediated internalization	Cav-1	[55]
*Neisseria gonorrhoeae*	Caveolin-mediated invasion	Cav-1	[90,91]
*Porphyromonas gingivalis*	Caveolin-mediated internalization	Cav-1	[89]
*Pseudomonas aeruginosa*	Lipid raft-mediated endocytosis	Cav-1 and Cav-2	[25,94,95]
*Rickettsia* spp.	Caveolin-mediated endocytic pathway for bacterial entry	Cav-1 and Cav-2	[95,98,99]
*Salmonella enterica* serovar Typhimurium	Caveolin-mediated internalization and transcytosis	Cav-1 and Cav-2.	[102,104]
*Streptococcus* spp.	Invasion and intracellular survival; caveosome-mediated internalization	Cav-1	[56]
Intracellular survival and cell–cell spread			
*Leptospira*	Intracellular migration through the vesicular transport system initiated by caveolin	Cav-1	[87]
*Listeria monocytogenes*	Cell-to-cell spreading	Cav-1	[50]
*Shigella flexneri*	Cell-to-cell spreading	Cav-1	[106]
Modulation of host immune responses			
*Escherichia coli* K1	Increase inflammation in brain cells	Cav-1	[80]
*Klebsiella pneumoniae*	Modulation of host immunity through STAT5-Akt signaling pathway	Cav-1	[86]
*Mycobacterium bovis* Bacillus Calmette-Guérin (BCG)	Cav-1 regulates apoptosis and the inflammatory response in macrophages infected with BCG	Cav-1	[53]
*Pseudomonas aeruginosa*	Downregulates inflammatory response in host cells	Cav-1	[94,96,97]
*Salmonella enterica* serovar Typhimurium	Regulate anti-inflammatory responses in macrophages	Cav-1	[105]

## Data Availability

This is a review article, and all data analyzed or discussed are publicly available in the references cited within the manuscript. No new datasets were generated or analyzed.

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
