# Peer review of "Caveolin-Mediated Endocytosis: Bacterial Pathogen Exploitation and Host–Pathogen Interaction"

_cells, 2024, doi:10.3390/cells14010002_

Round 1
Reviewer 1 Report
Comments and Suggestions for Authors
This is a very nice manuscript covering the role of caveolins in a range of bacterial pathogens with intracellular lifestyles. The authors have provided an interesting and detailed discussion of the topic. I have only a few minor suggestions to improve the manuscript:
-
Formatting in Section 2.1
- The authors should review the italicization of bacterial species names in Section 2.1 to ensure consistency and adherence to scientific formatting standards.
-
Figure 2
- It would be helpful to know if the authors have obtained the necessary approval for reproducing this figure. If the image is adapted or reused from another source, appropriate permissions and credits should be confirmed and clearly indicated.
-
Content Expansion
- While the manuscript covers a range of bacterial pathogens, it could benefit from the inclusion of additional information on the roles of caveolins in the infection mechanisms of:
- Helicobacter pylori: More details about how caveolins contribute to H. pylori pathogenesis and intracellular persistence would enhance this review.
- Coxiella burnetii and Mycoplasma spp.: These are important intracellular pathogens that exploit caveolae or caveolin-rich domains for their internalization and survival. Expanding the discussion to include these bacteria would provide a more comprehensive overview of the field.
- While the manuscript covers a range of bacterial pathogens, it could benefit from the inclusion of additional information on the roles of caveolins in the infection mechanisms of:
Overall, this is a strong and well-written review that will be of great interest to researchers studying bacterial pathogenesis and host-pathogen interactions. I look forward to seeing these minor revisions incorporated into the final version.
Reviewer 2 Report
Comments and Suggestions for Authors
This is a well-written and well organized review manuscript aiming to provide a comprehensive view on the role of caveolins and caveolin-mediated internalization in the cellular infection by several human bacterial pathogens.
This review is timely and will certainly be of the interest of the vast community working on host-pathogen interactions.
Minor points:
1) Figure 2 is not necessary. Should be removed.
2) In lines 172-178 page 5 the authors may also cite Gessain et al 2015, J Exp Med.
3) concerning Figure 3, the authors need to indicate what is shown in the left and right sides, and start the description from left to right. Phagocytic cell (left) and non-phagocytic cell (right). The legend should start be describing what happens in phagocytic cell (left).
Reviewer 3 Report
Comments and Suggestions for Authors
The review is outstanding and comprehensive. It can be accepted at its current form. It is very informative for people interesting in bacterial pathogens. My only suggestion is to have more information on how the pathogens are released into the cytoplasm once they enter the cells via endocytosis.
